# Inverse Kernel Decomposition

**Chengrui Li**  *cnlichengrui@gatech.edu*
*School of Computational Science & Engineering*
*Georgia Institute of Technology*

**Anqi Wu**  *anqiwu@gatech.com*
*School of Computational Science & Engineering*
*Georgia Institute of Technology*

**Reviewed on OpenReview:** *https://openreview.net/forum?id=H4OE7toXpa*

## Abstract

The state-of-the-art dimensionality reduction approaches largely rely on complicated optimization procedures. On the other hand, closed-form approaches requiring merely eigen-decomposition do not have enough sophistication and nonlinearity. In this paper, we propose a novel nonlinear dimensionality reduction method—Inverse Kernel Decomposition (IKD)—based on an eigen-decomposition of the sample covariance matrix of data. The method is inspired by Gaussian process latent variable models (GPLVMs) and has comparable performance with GPLVMs. To deal with very noisy data with weak correlations, we propose two solutions—blockwise and geodesic—to make use of locally correlated data points and provide better and numerically more stable latent estimations. We use synthetic datasets and four real-world datasets to show that IKD is a better dimensionality reduction method than other eigen-decomposition-based methods, and achieves comparable performance against optimization-based methods with faster running speeds. Open-source IKD implementation in Python can be accessed at `https://github.com/JerrySoybean/ikd`.

## 1 Introduction

Dimensionality reduction techniques have been widely studied in the machine learning field for many years, with massive applications in latent estimation (Wu et al., 2017; 2018), noise reduction (Sheybani & Javidi, 2009), cluster analysis (Bakrania et al., 2020), data visualization (Van der Maaten & Hinton, 2008a) and so forth. The most commonly used method is the principled component analysis (PCA), a linear dimensionality reduction approach. It is favored thanks to the easy use of a one-step eigen-decomposition. Its simple linear assumption, however, restricts its exploitation, especially in highly nonlinear scenarios. On the other hand, nonlinear dimensionality reduction models, such as autoencoders (Kramer, 1991), variational autoencoders (VAE) (Kingma & Welling, 2013), t-SNE (Van der Maaten & Hinton, 2008b), UMAP (McInnes et al., 2020), and Gaussian process latent variable models (GPLVMs) (Lawrence, 2003; 2005) can achieve state-of-the-art (SOTA) performance in terms of finding (sub)optimal low-dimensional latent and rendering satisfactory downstream analyses (e.g., visualization, prediction, classification). However, all these nonlinear models involve intricate optimization which is time-consuming, easy to get stuck in bad local optima, and sensitive to initialization. In this paper, we propose a novel nonlinear eigen-decomposition-based dimensionality reduction approach that finds low-dimensional latent with a closed-form solution but intricate nonlinearity.

The proposed method is called Inverse Kernel Decomposition (IKD), inspired by GPLVMs. GPLVMs are probabilistic dimensionality reduction generative models that use Gaussian Processes (GPs) to find a lower dimensional nonlinear embedding of high-dimensional data. GPLVM and its many variants have been proposed in various domains (Bui & Turner, 2015; Wang et al., 2005; 2008; Urtasun et al., 2006; Wu et al., 2017) and proven to be powerful nonlinear dimensionality reduction and latent variable models. However, GPLVMs are highly nonlinear and non-convex due to the GP component, resulting in practical difficulties

during optimization. By deriving the relationship implied by the kernel function in GPLVMs, IKD solves the GPLVM problem through eigen-decomposition, which could give a more stable latent estimation in a shorter time than the traditional optimization-based GPLVM solver (Sec. 3.1.1).

In the experiment section, we compare IKD against four eigen-decomposition-based and four optimization-based dimensionality reduction methods using synthetic datasets and four real-world datasets, and we can summarize four contributions of IKD:

• As an eigen-decomposition-based method, IKD achieves more reasonable latent representations than other eigen-decomposition-based methods with better classification accuracy in downstream classification tasks. The running time of IKD is on par with other eigen-decomposition-based methods.

• IKD is able to provide competitive performance against some SOTA optimization-based methods but at a much faster running speed.

• IKD promises a stable and unique optimal solution up to an affine transformation. In contrast, optimization-based methods do not guarantee unique optimal solutions and sometimes are not numerically stable due to the highly nonconvex optimization landscapes. Therefore, IKD can sometimes achieve better latent representations and classification performance than optimization-based methods like GPLVM and VAE.

• When the observation dimensionality is large (i.e. observation data is high-dimensional), a lot of methods have significant drawbacks. For example, t-SNE, UMAP, and VAE encounter the curse of dimensionality problems. The large dimensionality not only leads to longer running time but also hurts the dimensionality reduction performance. In contrast, IKD always obtains improved performance with an increasing observation dimensionality, and claims its absolute superiority when the observation dimensionality is very large.

Note that we are not claiming to propose the best dimensionality reduction approach that beats all other SOTAs. We propose an advanced eigen-decomposition-based method that (1) outperforms other eigen-decomposition-based methods in most synthetic and real-world applications with the same scale of running speeds and (2) reaches a comparable level against other optimization-based methods but with much faster running speed.

## 1.1 Related works

As GPLVM's eigen-decomposition-based solver, IKD starts from the data-generating process instead of the observed data. Although IKD makes use of eigen-decomposition and kernel functions, it is different from other eigen-decomposition-based methods. For example, GPLVM uses a kernel function to form a nonlinear mapping from the embedded latent space to the data space, which is opposite to the use of kernel as in kernel PCA (Schölkopf et al., 1997). Another similar method is Isomap (Tenenbaum et al., 2000), which is a generalized version of multidimensional scaling (MDS) (Borg & Groenen, 2005). From the form of the backbone IKD algorithm derived in Sec. 2.2, IKD obtains its target similarity matrix to be decomposed from data's pairwise correlations, and Isomap obtains its target similarity matrix from data's pairwise generalized distances. However, the initial goals of IKD and Isomap are different. Isomap hopes to place the dataset into a low-dimensional space while preserving pairwise distances as well-equally scaled as possible, but IKD works as an eigen-decomposition versioned solver for data generalized from GPVLM. In Sec 2, we introduce the generative model GPLVM first, and then convert the problem to an eigen-decomposition problem step-by-step, and finally solve it.

## 2 Methodology

### 2.1 Gaussian process latent variable model

**Generative model.** Let $\boldsymbol{X} \in \mathbb{R}^{T \times N}$ be the observed data where $T$ is the number of observations and $N$ is the observation dimensionality of each data vector. Let $\boldsymbol{Z} \in \mathbb{R}^{T \times M}$ denote its associated latent variables where $M$ is the latent dimensionality. Usually, we assume the latent space is lower-dimensional than the original observational space, leading to $M < N$. For each dimension of $\boldsymbol{X}$ denoted as $\boldsymbol{X}_{:,n} \in \mathbb{R}^T$, $\forall n \in \{1, \ldots, N\}$, GPLVM defines a mapping function that maps the latent to the observation which has a Gaussian

process (GP) prior. Therefore, given the finite number of observations, we can write

$$\boldsymbol{X}_{:,n} \overset{\text{i.i.d}}{\sim} \mathcal{N}(\boldsymbol{0}, \boldsymbol{K}), \quad \forall n \in \{1, 2, ..., N\}. \tag{1}$$

where $\boldsymbol{K}$ is a $T \times T$ covariance matrix generated by evaluating the kernel function $k$ of GP at all pairs of rows in $\boldsymbol{Z}$, i.e., $k_{i,j} = k(\boldsymbol{z}_i, \boldsymbol{z}_j)$ where $\boldsymbol{z}_i$ and $\boldsymbol{z}_j$ are the $i^{\text{th}}$ and $j^{\text{th}}$ rows of $\boldsymbol{Z}$.

**Problem setting.** The goal of GPLVM is to estimate the unknown latent variables $\boldsymbol{Z}$ that are used for constructing the covariance matrix $\boldsymbol{K}$, from the observations $\boldsymbol{X}$. Note that we only consider noiseless GP for the derivation of IKD, but IKD can deal with noisy observations empirically.

## 2.2 Inverse kernel decomposition

In this section, we derive a novel nonlinear decomposition method, inverse kernel decomposition (IKD), inspired by GPVLM. Previous work has been solving GPLVM by maximizing the log-likelihood to obtain $\boldsymbol{Z}$ from $\boldsymbol{X}$ in a one-step fashion. Now let us break this process into two steps: (1) estimating $\boldsymbol{K}$ from the observations $\boldsymbol{X}$, and (2) identifying the latent variables $\boldsymbol{Z}$ from the estimated covariance matrix $\boldsymbol{K}$. The first step can be solved by estimating $\boldsymbol{K}$ with the unbiased estimator, i.e., sample covariance $\boldsymbol{S} := \frac{1}{N-1}\left(\boldsymbol{X} - \bar{\boldsymbol{X}}\boldsymbol{1}^{\text{T}}\right)\left(\boldsymbol{X} - \bar{\boldsymbol{X}}\boldsymbol{1}^{\text{T}}\right)^{\text{T}} \approx \frac{1}{N-1}\boldsymbol{X}\boldsymbol{X}^{\text{T}}$, where $\bar{\boldsymbol{X}} = \frac{1}{N}\sum_{n=1}^{N}\boldsymbol{X}_{:,n}$ ought to be $\boldsymbol{0}$ since $\boldsymbol{X}_{:,n}$ are i.i.d. samples from a zero-mean Gaussian (Eq. 1).

Therefore, our main focus is to estimate the latent $\boldsymbol{Z}$ given $\boldsymbol{S}$ in the second step. In the following, we focus on the discussion of a commonly used stationary kernel, the squared exponential (SE) kernel. We will show in Sec. 2.3 that IKD can also work with various stationary kernels.

The SE kernel is defined as $k(\boldsymbol{z}_i, \boldsymbol{z}_j) = \sigma^2 \exp\left(-\frac{\|\boldsymbol{z}_i - \boldsymbol{z}_j\|^2}{2l^2}\right)$, where $\sigma^2$ is the marginal variance and $l$ is the length-scale. Note that $\sigma^2 = k(\boldsymbol{z}_i, \boldsymbol{z}_i) = k_{i,i}, \ \forall i \in \{1, \cdots, T\}$. Let $f$ be the scalar function mapping the scaled squared distance $d_{i,j} := \frac{\|\boldsymbol{z}_i - \boldsymbol{z}_j\|^2}{l^2}$ between latent $(\boldsymbol{z}_i, \boldsymbol{z}_j)$ to the scalar covariance $k_{i,j}$, i.e., $k_{i,j} = f(d_{i,j}) = \sigma^2 \exp\left(-\frac{d_{i,j}}{2}\right)$. Let us assume we know the true $\boldsymbol{K}$ for now. Since $f(\cdot)$ is strictly monotonic, we can obtain $d_{i,j} = f^{-1}(k_{i,j}) = -2\ln\left(\frac{k_{i,j}}{\sigma^2}\right)$. Writing $d_{i,j}$ in the matrix form $\boldsymbol{D} = (d_{i,j})_{T \times T}$, we have

$$\begin{aligned}
\boldsymbol{D} =& \frac{1}{l^2}\begin{bmatrix} 0 & (\boldsymbol{z}_1 - \boldsymbol{z}_2)^{\text{T}}(\boldsymbol{z}_1 - \boldsymbol{z}_2) & \cdots & (\boldsymbol{z}_1 - \boldsymbol{z}_T)^{\text{T}}(\boldsymbol{z}_1 - \boldsymbol{z}_T) \\ (\boldsymbol{z}_2 - \boldsymbol{z}_1)^{\text{T}}(\boldsymbol{z}_2 - \boldsymbol{z}_1) & 0 & \cdots & (\boldsymbol{z}_2 - \boldsymbol{z}_T)^{\text{T}}(\boldsymbol{z}_2 - \boldsymbol{z}_T) \\ \vdots & \vdots & \ddots & \vdots \\ (\boldsymbol{z}_T - \boldsymbol{z}_1)^{\text{T}}(\boldsymbol{z}_T - \boldsymbol{z}_1) & (\boldsymbol{z}_T - \boldsymbol{z}_2)^{\text{T}}(\boldsymbol{z}_T - \boldsymbol{z}_2) & \cdots & 0 \end{bmatrix} \\
=& f^{-1}(\boldsymbol{K}) = \left[f^{-1}(k_{i,j})\right]_{T \times T},
\end{aligned} \tag{2}$$

where $f^{-1}$ maps $\boldsymbol{K}$ to $\boldsymbol{D}$ element-wisely. We define $\tilde{\boldsymbol{z}} = \frac{\boldsymbol{z} - \boldsymbol{z}_1}{l}$ with $\tilde{\boldsymbol{z}}_1 = \boldsymbol{0}$. Now we have

$$\begin{aligned}
d_{i,j} =& \frac{1}{l^2}(\boldsymbol{z}_i - \boldsymbol{z}_j)^{\text{T}}(\boldsymbol{z}_i - \boldsymbol{z}_j) = \frac{1}{l^2}\left[(\boldsymbol{z}_i - \boldsymbol{z}_1) - (\boldsymbol{z}_j - \boldsymbol{z}_1)\right]^{\text{T}}\left[(\boldsymbol{z}_i - \boldsymbol{z}_1) - (\boldsymbol{z}_j - \boldsymbol{z}_1)\right] \\
=& (\tilde{\boldsymbol{z}}_i - \tilde{\boldsymbol{z}}_j)^{\text{T}}(\tilde{\boldsymbol{z}}_i - \tilde{\boldsymbol{z}}_j) = \tilde{\boldsymbol{z}}_i^{\text{T}}\tilde{\boldsymbol{z}}_i + \tilde{\boldsymbol{z}}_j^{\text{T}}\tilde{\boldsymbol{z}}_j - 2\tilde{\boldsymbol{z}}_i^{\text{T}}\tilde{\boldsymbol{z}}_j.
\end{aligned} \tag{3}$$

Since $\tilde{\boldsymbol{z}}_1 = \boldsymbol{0}$, we have $d_{1,j} = \tilde{\boldsymbol{z}}_1^{\text{T}}\tilde{\boldsymbol{z}}_1 + \tilde{\boldsymbol{z}}_j^{\text{T}}\tilde{\boldsymbol{z}}_j - 2\tilde{\boldsymbol{z}}_1^{\text{T}}\tilde{\boldsymbol{z}}_j \implies \tilde{\boldsymbol{z}}_j^{\text{T}}\tilde{\boldsymbol{z}}_j = d_{1,j}, \ \forall j \in \{1, ..., T\}$. Therefore, we arrive at an expression of $\tilde{\boldsymbol{z}}_i^{\text{T}}\tilde{\boldsymbol{z}}_j$ as $\tilde{\boldsymbol{z}}_i^{\text{T}}\tilde{\boldsymbol{z}}_j = \frac{1}{2}(d_{i,1} + d_{1,j} - d_{i,j})$. Note that $d_{1,i} = d_{i,1}$ because of the symmetric property. Denote $\tilde{\boldsymbol{Z}} = [\tilde{\boldsymbol{z}}_1, \tilde{\boldsymbol{z}}_2, \cdots, \tilde{\boldsymbol{z}}_T]^{\text{T}} = [\boldsymbol{0}, \tilde{\boldsymbol{z}}_2, \cdots, \tilde{\boldsymbol{z}}_T]^{\text{T}} \in \mathbb{R}^{T \times M}$, we could write the matrix form $\tilde{\boldsymbol{Z}}\tilde{\boldsymbol{Z}}^{\text{T}} = \left(\tilde{\boldsymbol{z}}_i^{\text{T}}\tilde{\boldsymbol{z}}_j\right)_{T \times T}$ as

$$\tilde{\boldsymbol{Z}}\tilde{\boldsymbol{Z}}^{\text{T}} = \begin{bmatrix} 0 & 0 & \cdots & 0 \\ 0 & d_{2,1} & \cdots & \frac{1}{2}(d_{2,1} + d_{1,T} - d_{2,T}) \\ 0 & \frac{1}{2}(d_{3,1} + d_{1,2} - d_{3,2}) & \cdots & \frac{1}{2}(d_{3,1} + d_{1,T} - d_{3,T}) \\ \vdots & \vdots & \ddots & \vdots \\ 0 & \frac{1}{2}(d_{T,1} + d_{1,2} - d_{T,2}) & \cdots & d_{T,1} \end{bmatrix} =: g(\boldsymbol{D}) = g(f^{-1}(\boldsymbol{K})), \tag{4}$$

which is a rank-$M$ symmetric positive semi-definite matrix given $M < T$. $g$ is the function mapping $\boldsymbol{D}$ to $\tilde{\boldsymbol{Z}}\tilde{\boldsymbol{Z}}^{\mathrm{T}}$. Then, Eq. 4 has the unique "reduced" eigen-decomposition

$$g(f^{-1}(\boldsymbol{K})) = \boldsymbol{U}\boldsymbol{\Lambda}\boldsymbol{U}^{\mathrm{T}} = \left(\sqrt{\lambda_1}\boldsymbol{U}_{:,1}, \ldots, \sqrt{\lambda_M}\boldsymbol{U}_{:,M}\right)\left(\sqrt{\lambda_1}\boldsymbol{U}_{:,1}, \ldots, \sqrt{\lambda_M}\boldsymbol{U}_{:,M}\right)^{\mathrm{T}} =: \tilde{\boldsymbol{U}}\tilde{\boldsymbol{U}}^{\mathrm{T}}, \tag{5}$$

where $\boldsymbol{U}_{:,m} = [0, u_{2,m}, u_{3,m}, \ldots, u_{T,m}]^{\mathrm{T}} \in \mathbb{R}^T$ is the $m^{\mathrm{th}}$ column of $\boldsymbol{U} \in \mathbb{R}^{T \times M}$ and $\boldsymbol{\Lambda} = \mathrm{diag}(\lambda_1, \lambda_2, \ldots, \lambda_M)$ with $\lambda_1 > \lambda_2 > \cdots > \lambda_M > \lambda_{M+1} = \cdots = \lambda_T = 0$. Note that the unique "reduced" singular value decomposition of $\tilde{\boldsymbol{Z}}$ is

$$\tilde{\boldsymbol{Z}} = \boldsymbol{U}\boldsymbol{\Lambda}^{\frac{1}{2}}\boldsymbol{V}^{\mathrm{T}} = \tilde{\boldsymbol{U}}\boldsymbol{V}^{\mathrm{T}} \implies \boldsymbol{z}_t = l\boldsymbol{V}\tilde{\boldsymbol{U}}_{t,:}^{\mathrm{T}} + \boldsymbol{z}_1, \quad \forall t \in \{1, \ldots, T\}, \tag{6}$$

where $\boldsymbol{z}_1$ represents the reference translation, the length-scale $l$ is a scaling factor, and $\boldsymbol{V}$ is an orthogonal matrix that is responsible for the corresponding rotation and reflection. Since $\boldsymbol{Z}$ and $\tilde{\boldsymbol{U}}$ span the same column space such that

$$\begin{aligned} k(\boldsymbol{z}_i, \boldsymbol{z}_j) &= \sigma^2 \exp\left(-\frac{\|\boldsymbol{z}_i - \boldsymbol{z}_j\|^2}{2l^2}\right) = \sigma^2 \exp\left(-\frac{\|l\boldsymbol{V}\tilde{\boldsymbol{U}}_{i,:}^{\mathrm{T}} - l\boldsymbol{V}\tilde{\boldsymbol{U}}_{j,:}^{\mathrm{T}}\|^2}{2l^2}\right) \\ &= \sigma^2 \exp\left(-\frac{\|\tilde{\boldsymbol{U}}_{i,:} - \tilde{\boldsymbol{U}}_{j,:}\|^2}{2}\right) = k_{l=1}\left(\tilde{\boldsymbol{U}}_{i,:}, \tilde{\boldsymbol{U}}_{j,:}\right). \end{aligned} \tag{7}$$

$\tilde{\boldsymbol{U}}$ contains all of the low-dimensional information of $\boldsymbol{Z}$. Therefore, we consider $\tilde{\boldsymbol{U}}$ as an estimator of $\boldsymbol{Z}$. Eq. 5 and Eq. 6 summarize the inverse relationship from a GP kernel covariance matrix to the latent variable.

To date, we are able to find the exact estimation $\tilde{\boldsymbol{U}}$ given the true GP covariance kernel $\boldsymbol{K}$ that is constructed from $\boldsymbol{Z}$ (Eq. 7). In practice, we only have the sample covariance estimator $\boldsymbol{S}$, and neither rank-$M$ nor positive semi-definite is guaranteed for $g(f^{-1}(\boldsymbol{S}))$. Therefore, we try to find its optimal rank-$M$ positive semi-definite approximation, i.e.

$$\underset{\tilde{\boldsymbol{U}} \in \mathbb{R}^{T \times M}}{\mathrm{minimize}} \left\|g(f^{-1}(\boldsymbol{S})) - \tilde{\boldsymbol{U}}\tilde{\boldsymbol{U}}^{\mathrm{T}}\right\|. \tag{8}$$

Dax et al. (2014) shows that

$$\tilde{\boldsymbol{U}} = \left(\sqrt{\lambda_1}\boldsymbol{U}_{:,1}, \ldots, \sqrt{\lambda_M}\boldsymbol{U}_{:,M}\right) \tag{9}$$

is the optimal solution for any unitarily invariant matrix norm $\|\cdot\|$, where $\lambda_1, \ldots, \lambda_M$ are the first $M$ largest positive eigenvalues of $g(f^{-1}(\boldsymbol{S}))$ and $\boldsymbol{U}_{:,1}, \ldots, \boldsymbol{U}_{:,M}$ are the corresponding eigenvectors. The goodness of the estimated latent via the target loss function Eq. 8 can be quantified by the explained variance ratio $\frac{\sum_{t=1}^M \lambda_t^2}{\sum_{t=1}^T \lambda_t^2}$. We summarize the IKD algorithm in Alg. 1.

---

**Algorithm 1** Inverse kernel decomposition

1: **function** IKD($\boldsymbol{X} \in \mathbb{R}^{T \times N}, f$)
2: $\quad \boldsymbol{S} \leftarrow \frac{1}{N-1}\left(\boldsymbol{X} - \bar{\boldsymbol{X}}\mathbf{1}^{\mathrm{T}}\right)\left(\boldsymbol{X} - \bar{\boldsymbol{X}}\mathbf{1}^{\mathrm{T}}\right)^{\mathrm{T}}$ $\qquad\qquad$ ▷ $\boldsymbol{S}$ serves as an estimator of the covariance $\boldsymbol{K}$
3: $\quad \sigma^2 \leftarrow \frac{1}{T}\sum_{i=1}^T s_{i,i}$ $\qquad\qquad$ ▷ estimate $\sigma^2$ through a statistic of the diagonal of $\boldsymbol{S}$
4: $\quad \hat{\boldsymbol{D}} = (\hat{d}_{i,j})_{T \times T} \leftarrow f^{-1}(\boldsymbol{S})$ $\qquad\qquad$ ▷ $\hat{\boldsymbol{D}}$ serves as an estimation of $\boldsymbol{D}$ (Eq. 2)
5: $\quad \boldsymbol{U}, \boldsymbol{\Lambda} \leftarrow$ eigen-decomposition of $g(\hat{\boldsymbol{D}})$ $\qquad\qquad$ ▷ Eq. 5
6: $\quad$ Form the optimal latent solution $\tilde{\boldsymbol{U}}$ using $\boldsymbol{U}$ and $\boldsymbol{\Lambda}$ $\qquad\qquad$ ▷ Eq. 5
7: $\quad$ **return** $\tilde{\boldsymbol{U}}$
8: **end function**

---

### 2.3 IKD with general stationary kernels

Apart from the SE kernel, IKD also works for most commonly used stationary kernels, as long as the kernel function $f$ is invertible (i.e., $f$ is strictly monotonic over $[0, \infty)$) and we can find a unique non-negative solution for $d = f^{-1}(k)$. We summarize the kernels in Tab. 1.

Table 1: Stationary kernels that can be applied to IKD.

| kernel | $f$ | $f^{-1}$ |
|---|---|---|
| squared exponential | $f(d) = \sigma^2 \exp\left(-\frac{d}{2}\right)$ | $f^{-1}(k) = -2\ln\left(\frac{k}{\sigma^2}\right)$ |
| rational quadratic | $f(d) = \sigma^2 \left(1 + \frac{d}{2\alpha}\right)^{-\alpha}$ | $f^{-1}(k) = 2\alpha\left[\left(\frac{k}{\sigma^2}\right)^{-\frac{1}{\alpha}} - 1\right]$ |
| $\gamma$-exponential | $f(d) = \sigma^2 \exp\left(-d^{\frac{\gamma}{2}}\right)$ | $f^{-1}(k) = \left(-\ln\frac{k}{\sigma^2}\right)^{\frac{2}{\gamma}}$ |
| Matérn | $f(d) = \sigma^2 \frac{2^{1-\nu}}{\Gamma(\nu)}\left(\sqrt{2\nu}\sqrt{d}\right)^\nu K_\nu\left(\sqrt{2\nu}\sqrt{d}\right)$ | no closed-form but solvable with root-finding algorithms |

For the SE kernel, we can generalize it to the ARD kernel $k(\boldsymbol{z}_i, \boldsymbol{z}_j) = \sigma^2 \exp\left(-\frac{1}{2}\sum_{m=1}^{M}\frac{1}{l_m^2}(z_{i,m} - z_{j,m})^2\right)$ and the Gaussian kernel $k(\boldsymbol{z}_i, \boldsymbol{z}_j) = \sigma^2 \exp\left(-\frac{1}{2}(\boldsymbol{z}_i - \boldsymbol{z}_j)^{\mathrm{T}}\boldsymbol{L}^{-1}(\boldsymbol{z}_i - \boldsymbol{z}_j)\right)$, where an extra affine transformation $\boldsymbol{L}^{\frac{1}{2}}$ is needed, rather than a constant scaling $l$.

For the Matérn kernel parameterized by $\nu$, $K_\nu(\cdot)$ is the modified Bessel function of the second kind. Although it is complicated to obtain a closed-form of $f^{-1}(\cdot)$ for the Matérn kernel, $f^{-1}(\cdot)$ always exists since $f(\cdot)$ is strictly monotonically decreasing over $[0, \infty)$ for all $\nu > 0$. Note that for the commonly used $\nu = p + \frac{1}{2}$, $p \in \mathbb{N}$, it is easy to derive $f'(\cdot)$, e.g., when $\nu = \frac{3}{2}$, $f(d) = \sigma^2\left(1 + \frac{\sqrt{3}d}{l}\right)\exp\left(-\frac{\sqrt{3}d}{l}\right)$, and $f'(d) = -\frac{3d\sigma^2}{l^2}\exp\left(-\frac{\sqrt{3}d}{l}\right)$. In such cases, higher-order root-finding algorithms (e.g., Newton's method) can be used to solve $d = f^{-1}(k)$.

The intuition of IKD is to find a non-linear mapping that makes $\boldsymbol{x}_i, \boldsymbol{x}_j$ that are strongly correlated in observational space to be closely located with each other in the latent space. This is perfectly reflected by the shape of these stationary kernels, i.e., monotonically decreasing fast near $d = 0$ and then becoming flat (converge to 0) when $d \to \infty$. Given these stationary kernels have such a similar shape, the latents estimated by different kernels are also similar (see the last paragraph of the real-world experiment, Sec. 3.2). Since these kernels are all invertible, we can derive the exact relationship between latents $\tilde{\boldsymbol{U}}^{(1)}$ and $\tilde{\boldsymbol{U}}^{(2)}$ from two kernels $f_1$ and $f_2$ respectively. Denote $\tilde{D}^{(1)} = \left[(\tilde{U}_{i,:}^{(1)} - \tilde{U}_{j,:}^{(1)})^{\mathrm{T}}(\tilde{U}_{i,:}^{(1)} - \tilde{U}_{j,:}^{(1)})\right]_{T\times T}$ and similar for $\tilde{D}^{(2)}$, then

$$f_1\left(\tilde{D}^{(1)}\right) = \boldsymbol{S} = f_2\left(\tilde{D}^{(2)}\right). \tag{10}$$

So, the transformation from $\tilde{D}^{(1)}$ to $\tilde{D}^{(2)}$ is the diffeomorphism $f_2^{-1}f_1$.

## 2.4 Error analysis of IKD

IKD performs eigen-decomposition on $g(f^{-1}(\boldsymbol{S}))$ (Eq. 5), which uses the sample covariance $\boldsymbol{S}$ as an empirical estimator of $\boldsymbol{K}$. In practice, sample covariance values $s_{i,j}$ in $\boldsymbol{S}$ can be very noisy due to the noise in the data and an insufficient observation dimensionality $N$. There can be non-positive and close-to-zero positive covariance values preventing from calculating $\hat{d}_{i,j} = f^{-1}(s_{i,j})$ accurately. Non-positive $s_{i,j}$ falls out of the input range of $f^{-1}$, i.e., $(0, \sigma^2]$. For close-to-zero positive $s_{i,j}$, the error between the estimation $\hat{d}_{i,j} = f^{-1}(s_{i,j})$ and the ground truth $d_{i,j} = f^{-1}(k_{i,j})$ can be large and sensitive to $s_{i,j}$. A sketch analysis of the error for the SE kernel is via the Taylor expansion of $f^{-1}$ at $s_{i,j}$:

$$
\begin{aligned}
d_{i,j} &= f^{-1}(k_{i,j}) = -2\ln\frac{s_{i,j} + (k_{i,j} - s_{i,j})}{\sigma^2} = -2\ln\frac{s_{i,j}}{\sigma^2} - 2\frac{k_{i,j} - s_{i,j}}{s_{i,j}} + O((k_{i,j} - s_{i,j})^2) \\
&= f^{-1}(s_{i,j}) + \frac{O(k_{i,j} - s_{i,j})}{s_{i,j}} = \hat{d}_{i,j} + \frac{O(k_{i,j} - s_{i,j})}{s_{i,j}}.
\end{aligned}
\tag{11}
$$

We define the estimation error as $|d_{i,j} - \hat{d}_{i,j}| = \frac{O(|k_{i,j} - s_{i,j}|)}{s_{i,j}}$. For large $s_{i,j}$, the error is small; but for small $s_{i,j}$, the error is very sensitive to the covariance error $|k_{i,j} - s_{i,j}|$. To resolve the issue, there are two solutions:

**Blockwise solution.** We first throw away bad $s_{i,j}$ values by thresholding the sample covariance with a value $s_0$, leading to a thresholded covariance matrix $\tilde{\boldsymbol{S}} = (s_{i,j} \cdot \mathbb{1}[s_{i,j} > s_0])_{T\times T}$. $\tilde{\boldsymbol{S}}$ is not a fully connected

graph due to the zero values. We can not directly apply IKD to $\tilde{\boldsymbol{S}}$ for latent estimation. Then, we can use, for example, the Bron–Kerbosch algorithm (Bron & Kerbosch, 1973), to find maximal cliques in $\tilde{\boldsymbol{S}}$. Consequently, each clique is a fully connected subgraph (block) of $\tilde{\boldsymbol{S}}$, which can be decomposed using IKD. After obtaining the latent for each clique, we merge all of the estimated latent variables according to the shared points between every clique pair. As long as the number of shared points between two cliques is greater than $M$, we are able to find the unique optimal rigid transformation that aligns every two cliques correctly.

We clarify the details of the clique merging procedure in the blockwise solution here. First, the Bron-Kerbosch algorithm provides us with a set of cliques $\{C_1, C_2, \ldots, C_L\}$ satisfying $\bigcup_{i=1}^{L} C_i = \{\boldsymbol{z}_t\}_{t=1}^{T}$. Now, we start with two cliques that share the maximum number of points, $\arg\max_{C_i, C_j} |C_i \cap C_j|$. For example, now we have $C_i = \{\boldsymbol{z}_1, \boldsymbol{z}_2, \boldsymbol{z}_3, \boldsymbol{z}_4\}$ and $C_2 = \{\boldsymbol{z}_2, \boldsymbol{z}_3, \boldsymbol{z}_4, \boldsymbol{z}_5\}$ (Fig 1). Their shared points are $\{\boldsymbol{z}_2, \boldsymbol{z}_3, \boldsymbol{z}_4\}$. Since the latent is in $\mathbb{R}^M = \mathbb{R}^2$ and $|C_i \cap C_j| = 3 \geqslant M + 1$, we can find a unique reflection + rotation + translation via SVD to align $\{\boldsymbol{z}_2, \boldsymbol{z}_3, \boldsymbol{z}_4\}$ in $C_2$ with $\{\boldsymbol{z}_2, \boldsymbol{z}_3, \boldsymbol{z}_4\}$ in clique $C_1$. If there are less than $M + 1$ shared points between the two cliques, there is more than one way to align them. Similarly, in $\mathbb{R}^M$, the uniqueness of this optimal alignment transformation requires at least $M + 1$ shared points. Once two cliques are merged, we remove $C_i, C_j$ from the original set and place the merged cliques $C_i \cap C_j$ back. The final latent can be obtained by repeating this merging procedure until the largest clique contains all points.

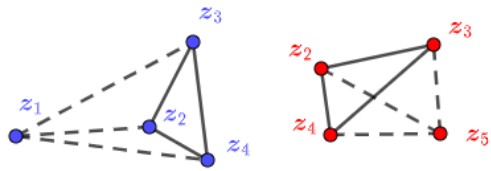

Figure 1: A clique pair example that shares three points $\{\boldsymbol{z}_2, \boldsymbol{z}_3, \boldsymbol{z}_4\}$.

Although the complexity of the Bron-Kerbosch algorithm for finding maximal cliques is $\mathcal{O}(3^T)$, we can terminate the algorithm as long as the union of the existing cliques is the whole dataset. In other words, we only need up to $T$ maximal cliques, so the clique finding time can be bounded by $\mathcal{O}(T^2)$. Then solving first $M$ eigen-decomposition algorithm for up to $T$ cliques takes $\mathcal{O}(T \times (MT^2))$. Therefore, the complexity of the entire procedure can be bounded by $\mathcal{O}(MT^3)$.

**Geodesic solution.** Since small values $s_{i,j} < s_0$ have significantly bad effects on eigen-decomposition, we can replace $s_{i,j}$, whose value is smaller than $s_0$, with the geodesic covariance $s_{i,j} \leftarrow \max_{(t_1, t_2, \ldots, t')} s_{i, t_1} \cdot s_{t_1, t_2} \cdots s_{t', j}$, where $i \to t_1 \to \cdots \to t' \to j$ is the geodesic path from $i$ to $j$ found by the Dijkstra algorithm (Dijkstra et al., 1959). The complexity of this approach is bounded by the complexity of the Dijkstra algorithm, which is $\mathcal{O}(T^2 \log(T))$. Since the complexity of the geodesic approach is smaller than that of the blockwise approach when $T$ is larger (greater than 1000 in the following experiments), we choose the geodesic instead of the blockwise. A comparison of these two solutions is shown in the experiment section.

## 2.5 Reference point selection

In Eq. 3, we choose $\boldsymbol{z}_1$ as the reference point to calculate $d_{i,j}$. But the reference point can be any of the points in $\{\boldsymbol{z}_t\}_{t=1}^{T}$. If we choose $\boldsymbol{z}_r$, for an arbitrary index $r \in \{1, ..., T\}$, to be the reference point, then similar to Eq. 4, the $r^{\text{th}}$ row and the $r^{\text{th}}$ column of $g(\boldsymbol{D})$ are 0s, and the remaining elements are $g(\boldsymbol{D})_{i,j} = \frac{1}{2}(d_{i,r} + d_{r,j} - d_{i,j}) \neq 0$, $\forall i \neq r, j \neq r$. Note that every $g(\boldsymbol{D})_{i,j}$ includes an element from $\{d_{r,i}\}_{i=1}^{T}$. Thus the quality of $\{d_{r,i}\}_{i=1}^{T}$ is vital for latent estimation. Based on the analysis in Eq. 11, we know that in practice we want to choose a good reference index $r$ so that $\{\hat{d}_{r,i}\}_{i=1}^{T}$ are relatively small (i.e., $\{s_{r,i}\}_{i=1}^{T}$ are large, which means the $r^{\text{th}}$ data point is highly correlated with the rest of the data points). Note that multidimensional scaling (MDS) (Kruskal & Wish, 1978) solves a similar eigen-decomposition problem, i.e., finding coordinates $\boldsymbol{Z}$ from the distance matrix $\boldsymbol{D}$. It employs a centering idea which is equal to using the average of all latent variables $\frac{1}{T}\sum_{t=1}^{T} \boldsymbol{z}_t$ as the reference point. We choose the best reference point instead since we want to reduce the estimation error in Eq. 11 as much as we can, so that the objective function in Eq. 8 can be

minimized as much as possible. Mathematically, we obtain $r$ such that $\|\hat{\boldsymbol{d}}_r\|_\infty \leqslant \|\hat{\boldsymbol{d}}_i\|_\infty$, $\forall i \in \{1, 2, ..., T\}$ and $\hat{\boldsymbol{d}}_i$ is the $i^{\text{th}}$ row of $\hat{\boldsymbol{D}} = (\hat{d}_{i,j})_{T \times T}$, i.e., $r = \arg\min_i \|\hat{\boldsymbol{d}}_i\|_\infty = \arg\min_i \left\{ \max_j \{\hat{d}_{i,j}\} \right\}$.

# 3 Experiments

In this section, we evaluate IKD with the most commonly used squared exponential as the default kernel on three synthetic datasets, where we know the true latent representations and four real-world datasets.

**Baseline methods for comparison**:
- **PCA**: Principal component analysis. One of the most widely used linear dimensionality reduction methods.
- **KPCA**: Kernel PCA. We try different kernels (polynomial, SE, sigmoid, and cosine) and present the best one.
- **LE** (Belkin & Niyogi, 2003): Laplacian eigenmaps, which is to do spectral decomposition to the affinity matrix's graph Laplacian. We use the nearest neighbors algorithm to build the affinity matrix.
- **Isomap** (Tenenbaum et al., 2000): Isometric mapping of multidimensional scaling (MDS) by incorporating the geodesic distances. We use `sklearn`'s default setting—five nearest neighbors—to build the distance matrix.
- **t-SNE** (Van der Maaten & Hinton, 2008b): $t$-distributed stochastic neighbor embedding. We use `sklearn`'s default hyperparameter setting to fit the model.
- **UMAP** (McInnes et al., 2020): Uniform Manifold Approximation and Projection for dimension reduction. We use `sklearn`'s default hyperparameter setting to fit the model. We use the official UMAP package (McInnes et al., 2018) and its default hyperparameter setting to fit the model.
- **GPLVM** (Lawrence, 2003; 2005): The traditional optimization-based Gaussian process latent variable model solver. We use the GPLVM module in the `GPy` package (GPy, since 2012) and its default hyperparameter setting to fit the model. Same as IKD, we use the most commonly used squared exponential as the default kernel unless otherwise stated.
- **VAE** (Kingma & Welling, 2013): Variational autoencoder.

The first four are eigen-decomposition-based methods; the last four are optimization-based methods.

## 3.1 Synthetic data

**Experimental setup.** We first test all methods on three synthetic datasets. All the following experiments are based on 50 independent repeats (trials). For each trial, we generate the true latent variables from

$$\boldsymbol{Z}_{m,1:T} \sim \mathcal{N}\left(\boldsymbol{0}, \left(6\mathrm{e}^{-\frac{|i-j|}{5}}\right)_{T \times T}\right), \quad \forall m \in \{1, ..., M\}, \tag{12}$$

where $M$ is the latent dimensionality, varying across different datasets. Then, we generate the noiseless data from GP, sinusoidal, and Gaussian bump mapping functions respectively. Afterward, i.i.d. Gaussian noise is added to form the final noisy observations $\boldsymbol{X}$.

**Evaluateion.** We evaluate the performance using the $R^2$ metric. When computing $R^2$ values, we first align the estimated latent with the ground truth through an affine transformation (i.e., a linear decoder); then compute $R^2$ for each latent dimension, and finally take an average across all latent dimensions $m \in \{1, 2, ..., M\}$. The reasons for choosing the affine transformation are: (1) rigid transformation could lead to very negative $R^2$ values for those non-IKD methods (e.g., PCA), not shown here; and (2) affine transformation is the commonly used one for latent estimation and alignment.

**Dataset 1: GP mapping function.** We start our experiments with the GP mapping function. In each trial, we generate a 3D latent $\boldsymbol{Z} \in \mathbb{R}^{1000 \times 3}$ (i.e., $M = 3$) according to Eq. 12, and generate $\boldsymbol{X} \in \mathbb{R}^{1000 \times N}$ according to Eq. 1 with $\sigma^2 = 1$ and $l = 3$. Then Gaussian noise is added: $x_{t,n} \leftarrow x_{t,n} + \varepsilon_{t,n}$, $\forall(t,n) \in \{1, ..., 1000\} \times \{1, ..., N\}$, where noise $\varepsilon_{t,n} \sim \mathcal{N}(0, 0.05^2)$. Note that this generating process is consistent with the generating process of GPLVM. Thus it is well aligned with the model assumptions of IKD, deemed as a

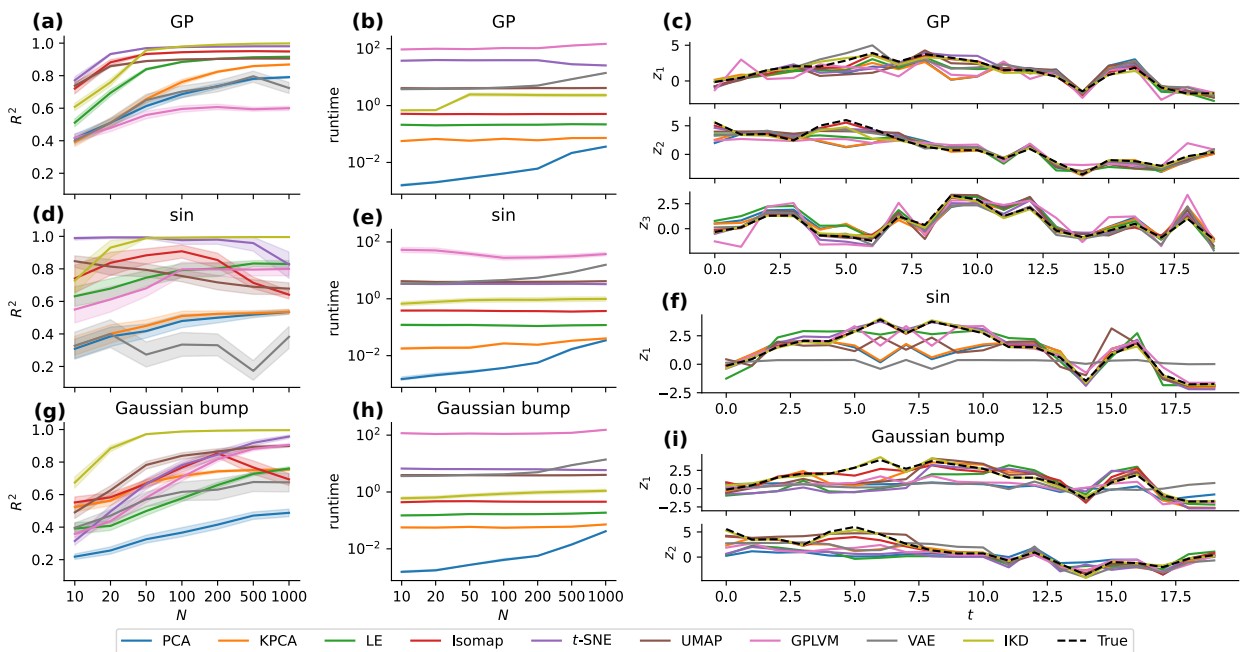

Figure 2: $R^2$ values (a,d,g) and running time (b,e,h) with respect to $N$ with GP, sinusoidal, and Gaussian bump mapping functions. (c,f,i): Latent recovery visualization of an example trial with $N = 100$, for the first 20 points with GP, sinusoidal, and Gaussian bump mapping functions.

data-matching example. Fig. 2(a) shows that for $N = 10$ and $N = 20$, Isomap is the best; but when $N > 50$, IKD becomes the best and its $R^2$ converges to 1 as $N$ increases. The latent recovery visualization of an example trial under $N = 100$ for the first 20 points (Fig. 2(c)) shows that Isomap and IKD match the true latent the best.

**Dataset 2: Sinusoidal mapping function.** In each trial, we generate a 1D latent (i.e., $M = 1$) according to Eq. 12, and generate the noisy observations $\boldsymbol{X} \in \mathbb{R}^{1000 \times N}$ as $\boldsymbol{x}_t = \sin(\boldsymbol{\Omega} \boldsymbol{z}_t + \boldsymbol{\varphi}) + \boldsymbol{\varepsilon}_t, \quad \forall t \in \{1, ..., 1000\}$, where $\boldsymbol{\Omega} = (\omega_{n,m})_{N \times M}$ with $\omega_{n,m} \sim \mathcal{U}(-1, 1)$, $\boldsymbol{\varphi} = [\varphi_1, ..., \varphi_N]^{\mathrm{T}}$ with $\varphi_n \sim \mathcal{U}(-\boldsymbol{\pi}, \boldsymbol{\pi})$, and noise $\boldsymbol{\varepsilon}_t \sim \mathcal{N}(\boldsymbol{0}, 0.1^2 \boldsymbol{I})$. The result in Fig. 2(d) indicates that even though the observed data is not from a GP (data-mismatching), IKD is still able to discover the latent structure consistently better than others, except for $N = 10$ and $N = 20$ where Isomap is the best. When the observation dimensionality $N > 50$, the $R^2$ value of IKD approaches to 1, while Isomap, t-SNE, and UMAP all have decreasing performance due to the curse of dimensionality. The latent recovery visualization of an example trial under $N = 100$ for the first 20 points (Fig. 2(f)) shows that Isomap and IKD match the true latent the best.

**Dataset 3: Gaussian bump mapping function.** In each trial, we generate a 2D latent (i.e., $M = 2$) according to Eq. 12, and generate $\boldsymbol{X} \in \mathbb{R}^{1000 \times N}$ as

$$x_{t,n} = 20 \exp \left( -\|\boldsymbol{z}_t - \boldsymbol{c}_n\|_2^2 \right) + \varepsilon_{t,n}, \quad \forall (t, n) \in \{1, ..., 1000\} \times \{1, ..., N\}, \tag{13}$$

where $\boldsymbol{c}_n \in \mathbb{R}^2$ is the center of the $n^{\mathrm{th}}$ Gaussian bump randomly selected from 10,000 grid points uniformly distributed in $[-6, 6]^2$, with noise $\varepsilon_{t,n} \sim \mathcal{N}(0, 0.05^2)$. This is another data-mismatching example. Fig. 2(g) shows that in this case, IKD is the best one among all methods for all observation dimensionality $N$. Fig. 2(i) also shows that only IKD matches the true latent accurately.

The running time of IKD in the three synthetic datasets above is on par with other eigen-decomposition-based methods (Fig.2(b,e,h)), and much less than optimization-based methods. In particular, GPLVM always takes a very long time; t-SNE requires more running time on datasets whose latent dimensionality is greater than

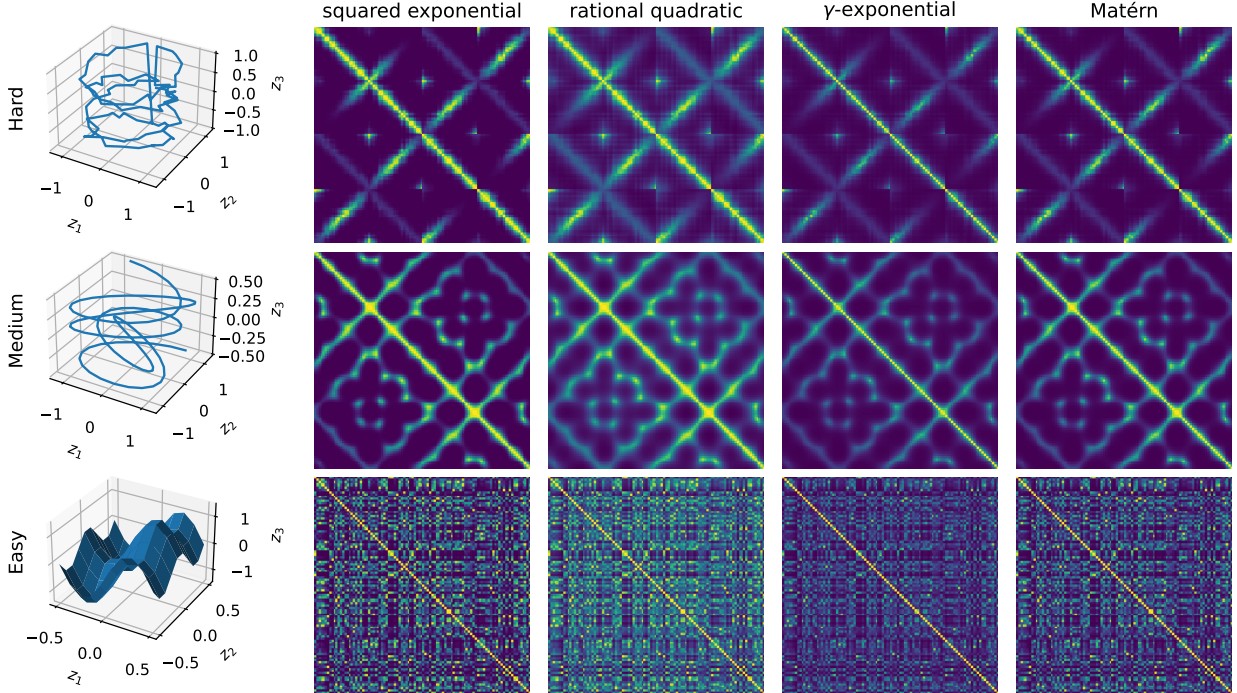

Figure 3: Left: ground truth $\mathbf{Z}$ of the latent. Right: kernel covariance matrix $\mathbf{K}$ created by the corresponding latent for the four different kernels, with marginal variance $\sigma = 1$ and length-scale $l = 0.5$; $\alpha = 1$ in rational quadratic kernel, $\gamma = 1$ in $\gamma$-exponential kernel, and $\nu = 1.5$ in Matérn kernel. The difficulty levels of the three datasets are from hard to easy, from top to bottom. For the easy dataset (bottom), points are selected from the grids on the surface and randomly permuted.

2; and VAE is more time-consuming when the observation dimensionality is large. Note that we only vary the dimensionality $N$ not the number of observations $T$. When increasing $T$, the running time of all methods will increase polynomially. For optimization-based methods, stochastic optimization can be employed to scale to large-scale datasets. For fair comparison, extra scaling techniques should be incorporated to deal with large-scale eigen-decomposition, which falls out of the scope of this paper.

In general, IKD performs the best for all three mapping functions especially when the observation dimensionality $N$ is large. It is also very effective in capturing details in addition to recovering the general latent structure correctly. Same as other eigen-decomposition-based methods, IKD takes less time to solve the presented problems compared with optimization-based methods.

### 3.1.1  Varying dimensionality, kernels, and latent structures

We test the effectiveness of IKD on the observation data generated from the GP mapping function described in Eq. 1, for different observation dimensionality $N \in \{100, 200, 500, 1000, 2000, 5000, 10000\}$, different generating kernels (Tab. 1), and three different latent structures (hard, medium, and easy shown in Fig. 3 according to their difficulty levels). We only compare IKD with PCA and GPLVM here. PCA is the most commonly used linear method, so it serves as a baseline. GPLVM is the traditional optimization-based solver, and IKD is our newly proposed eigen-decomposition-based solver for data generated from the GP mapping function.

From Fig. 4(b), we can tell that IKD is always the best for the most commonly used SE kernel. Compared with PCA and GPLVM, IKD is highly effective especially when $N$ is large, where the $R^2$ values of IKD are very close to 1. Fig. 4(c) shows the latent recovery visualization from one example trial of the medium dataset when $N = 1000$. The kernel of the generating model is SE. We can tell that IKD matches the ground

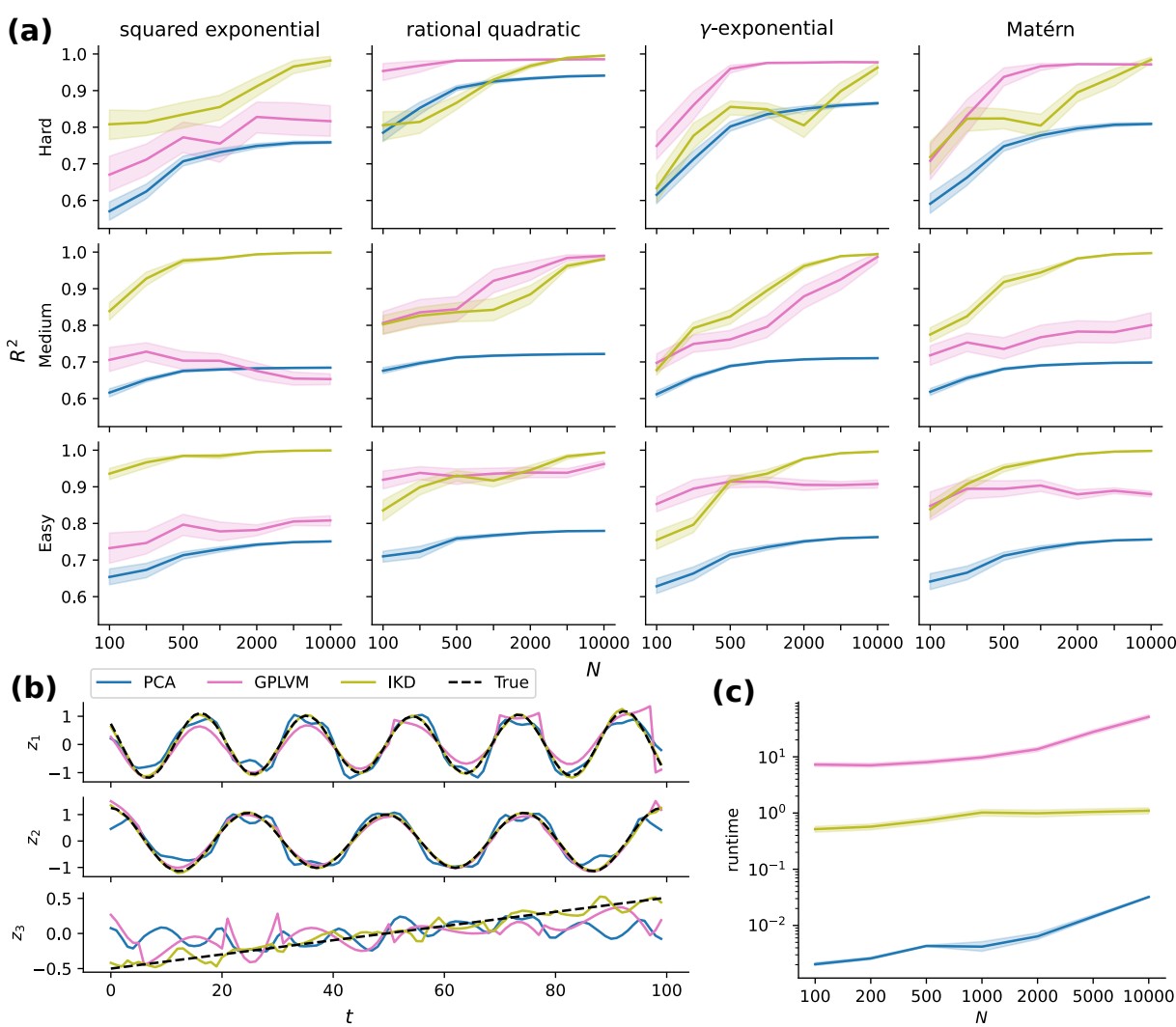

Figure 4: (a): $R^2$ values, with respect to $N$, of PCA, GPLVM, and IKD for different datasets and kernels. (b): Latent recovery visualization of an example trial of the medium dataset with the SE kernel and $N = 1000$. (c): Average running time in seconds (across different kernels, different datasets, and 50 independent trials) of the three methods w.r.t. observation dimensionality $N$. We take averages across different kernels and different datasets because all of them share similar running time results.

truth the best. For the third dimension, particularly, only the estimated latent from IKD reflects the linear increasing trend of the ground truth correctly. In terms of complexity, GPLVM is time-consuming compared with IKD (Fig. 4(d)). These results indicate that we can use IKD to recover the latent for data generated from the GP mapping function faster and more accurately.

### 3.1.2 Varying the noise level

To understand the performance of IKD solving GPLVM with different levels of noise, we use the medium dataset (Fig. 3) and the squared exponential kernel as the generative GPLVM to simulate the noiseless observation $\boldsymbol{X}$ by Eq. 1. Then, different leveled noises are added to the observation, i.e., $x_{t,n} \leftarrow x_{t,n} + \varepsilon_{t,n}$, where $\varepsilon_{t,n} \sim \mathcal{N}(0, \mathrm{sd}^2)$, for sd $\in \{0, 0.1, \ldots, 0.9, 1\}$. The standard deviation of the Gaussian noise "sd" represents the noise level.

Fig. 5 shows that IKD is able to have a relatively good estimation on datasets with low-level noise. The $R^2$ of IKD gradually decreases as the noise level increases, but it is still better than GPLVM up to the noise level of as high as sd $= 1$. The traditional optimization-based GPLVM solver explicitly considers the noise term in its algorithm and the performance even slightly improves as the noise increases. Besides, with larger noise, the observation also makes the GPLVM solver run faster.

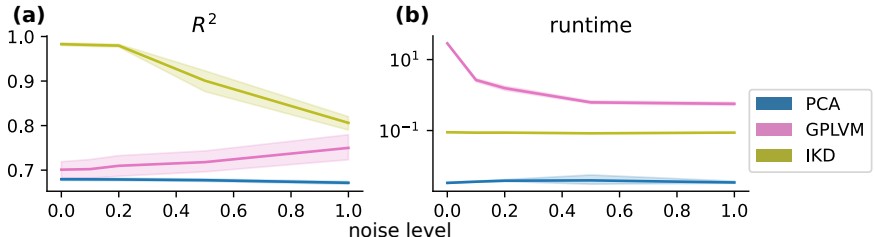

Figure 5: (a): $R^2$ performances of PCA, GPLVM, and IKD on datasets with different noise levels. (b): The corresponding running time.

### 3.1.3  Comparison of the blockwise and the geodesic solutions

To understand the differences between blockwise and geodesic solutions, we apply the blockwise and the geodesic IKD on the medium dataset (Fig. 3) respectively and visualize their results. There are two sets of observations, one is from the SE kernel and the other is from the rational quadratic kernel. We only use the SE kernel to solve both of the two observations, constituting a model match case and a model mismatch case.

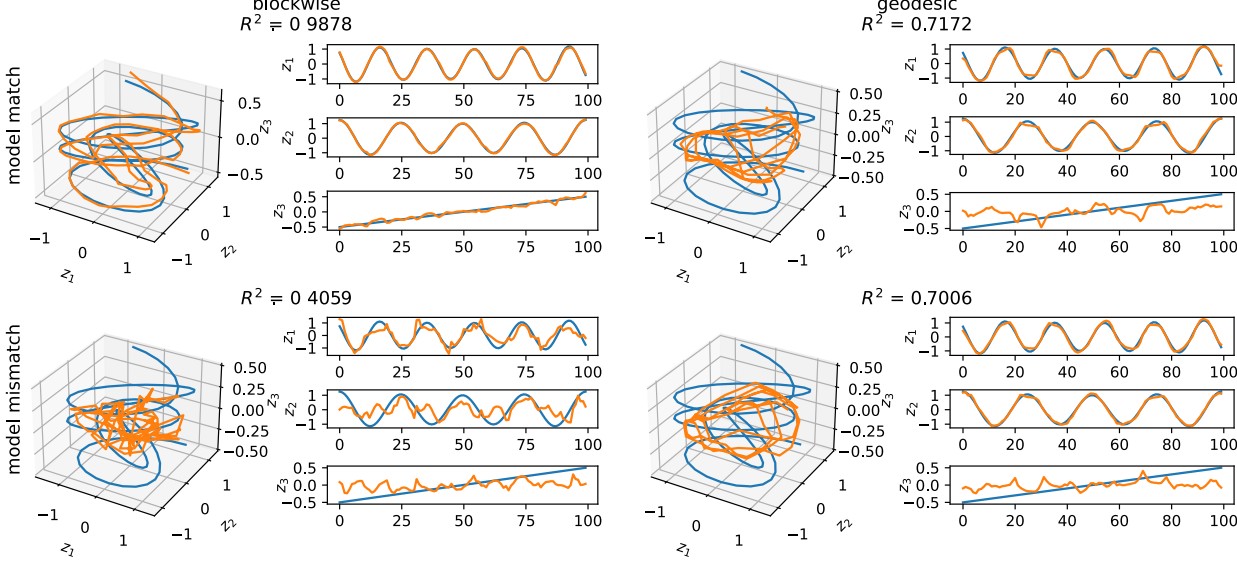

Figure 6: The blockwise (left) and geodesic (right) solutions in a model match case (above) and a model mismatch case (below). The blue curve is the true latent, and the orange curves are estimated latent. In the model mismatch case, although the true data-generating model is not GP with the SE kernel, we still use a simple decoder (i.e., an affine transformation) to align the estimated latent with the true latent to compute the $R^2$ metric.

In the model match case, the blockwise approach solves the GPLVM, but the geodesic approach does not. Therefore, in the model match case and when the number of data points is not large, the Bron-Kerbosch algorithm can be finished quickly and the blockwise solution should converge to the true latent up to an affine transformation (the model match case in Fig. 6).

In the model mismatch case, the clique-merging procedure is vulnerable to numerical errors, and small errors can accumulate during this procedure, especially when the number of cliques is large. And if the number of cliques is large but points are scattered, there will be too many cliques to be found and hence the running time of the Bron-Kerbosch algorithm is unacceptable. Moreover, we do not know whether the true data-generating model is GP with the SE kernel (and it is very likely that it is not GP, just like in PCA we know the true data-generating model is certainly not a simple linear model). In such cases, we treat all data points as a whole and compute their empirical correlation through the geodesic shortest path, and hence using geodesic is more intuitive, globally stable, and efficient (the model mismatch case in Fig. 6).

### 3.2 Real-world data

**Dataset.**  We compare IKD against alternatives on four real-world datasets:
• Single-cell qPCR (PRC) (Guo et al., 2010): Normalized measurements of 48 genes of a single cell at 10 different stages. There are 437 data points in total, resulting in $\boldsymbol{X} \in \mathbb{R}^{437 \times 48}$.
• Hand written digits (digits) (Dua & Graff, 2017): It consists 1797 grayscale images of hand written digits. Each one is an $8 \times 8$ image, resulting in $\boldsymbol{X} \in \mathbb{R}^{1797 \times 64}$.
• COIL-20 (Nene et al., 1996): It consists 1440 grayscale photos. For each one of the 20 objects in total, 72 photos were taken from different angles. Each one is a $128 \times 128$ image, resulting in $\boldsymbol{X} \in \mathbb{R}^{1440 \times 16384}$.
• Fashion MNIST (F-MNIST) (Xiao et al., 2017): It consists of 70000 grayscale images of 10 fashion items (clothing, bags, etc). We use a subset of it, resulting in $\boldsymbol{X} \in \mathbb{R}^{3000 \times 784}$.

**Evaluation.**  Since there is no true latent to compare against, we first estimate the latent in a $\{2, 3, 5, 10\}$-dimensional latent space and then use the $k$-nearest neighbor ($k$-NN) classifier to evaluate the performance of each dimensionality reduction method. Specifically, we apply 5-fold cross-validation $k$-NN ($k \in \{5, 10, 20\}$) on the estimated $\{2, 3, 5, 10\}$-dimensional latent to evaluate the performance of each method on each dataset. The $k$-NN classification results of different methods under different latent dimensionality $M$, different datasets, and different choices of $k$ are shown in Fig. 7(a). Performances of different methods on different datasets evaluated by the silhouette score are presented in Fig. 7(b).

**Results.**  Comparing IKD with other eigen-decomposition-based methods (PCA, KPCA, LE, Isomap), we can conclude that IKD is almost always the best one on all four datasets, except that when $M \in \{3, 5, 10\}$ in the digits dataset, Isomap is better than IKD. When comparing IKD with GPLVM, we find the performances of GPLVM on PCR, digits, and F-MNIST datasets are slightly better than IKD while GPLVM takes too much running time. Specifically, IKD is significantly better than GPLVM on the COIL-20 dataset but only slightly worse than GPLVM on the other three datasets. VAE only performs well on the most complicated dataset F-MNIST, and is much worse than IKD in the other three datasets. Although IKD is worse than the remaining two optimization-based methods (t-SNE and UMAP), the performance of IKD is the best on the COIL-20 dataset. The reason is that the observation dimensionality is very high ($N = 16384$) in the COIL-20 dataset, and IKD is very effective for high-dimensional data as shown in the synthetic results (Fig. 2(a)).

2D visualization of the four datasets are shown in Fig. 8, 9, 10, and 11 respectively. Qualitatively, we can see that IKD consistently finds more separate clusters compared with all other eigen-decomposition-based methods and two optimization-based methods (GPLVM and VAE) across all four datasets. Therefore, even though IKD is an eigen-decomposition-based method, its performance is significantly better than other eigen-decomposition methods on all four real-world datasets, sometimes as good as the best optimization-based method.

In terms of running time (Fig. 7(b)), IKD is on par with Isomap, and these eigen-decomposition-based methods are significantly faster than those four optimization-based methods. Note that if the desired latent dimensionality $M > 2$, the running time of t-SNE is barely acceptable. For the high dimensional COIL-20, the running time values of VAE and GPLVM are extremely high, getting out of the upper limit of the corresponding axes.

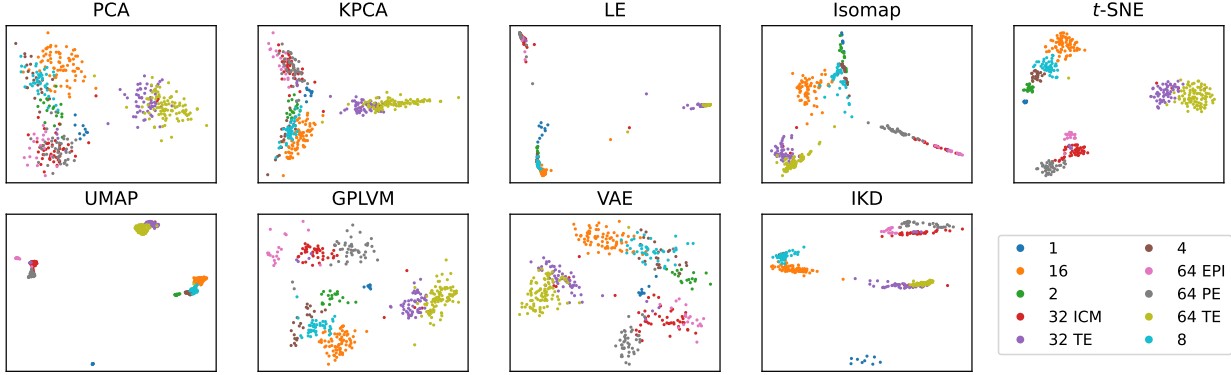

Figure 7: (a) $k$-NN 5-fold cross-validation, (b) silhouette score, and (c) running time, on different methods, different latent dimensionality $M$, different datasets, and different choices of $k$.

Figure 8: Visualization of the dimensionality reduction results of different methods on the PCR dataset.

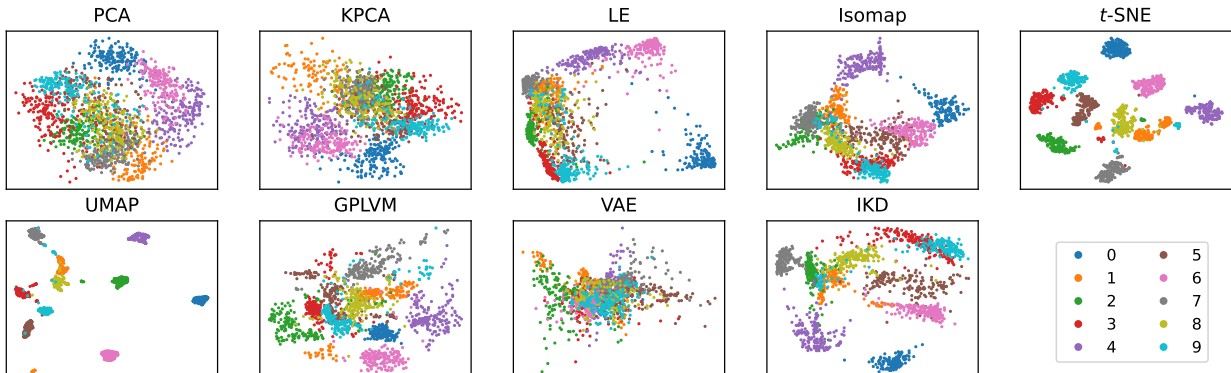

Figure 9: Visualization of the dimensionality reduction results of different methods on the digits dataset.

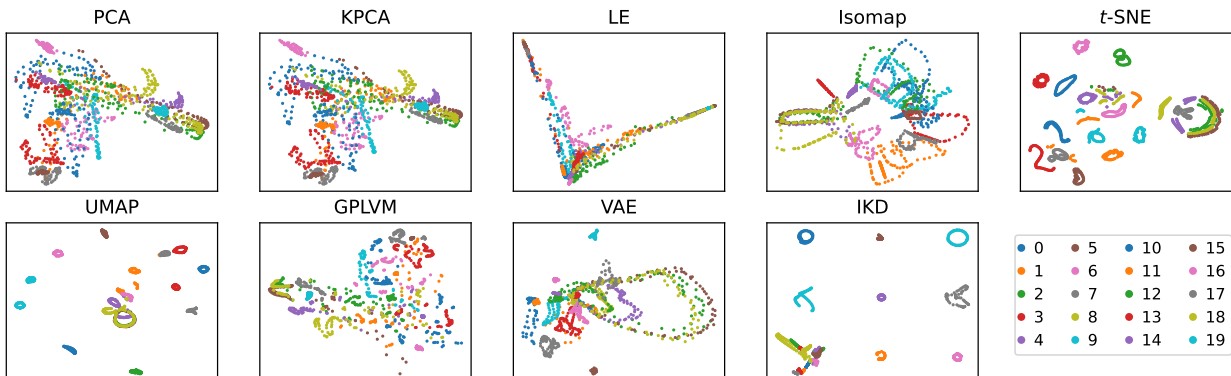

Figure 10: Visualization of the dimensionality reduction results of different methods on the COIL-20 dataset.

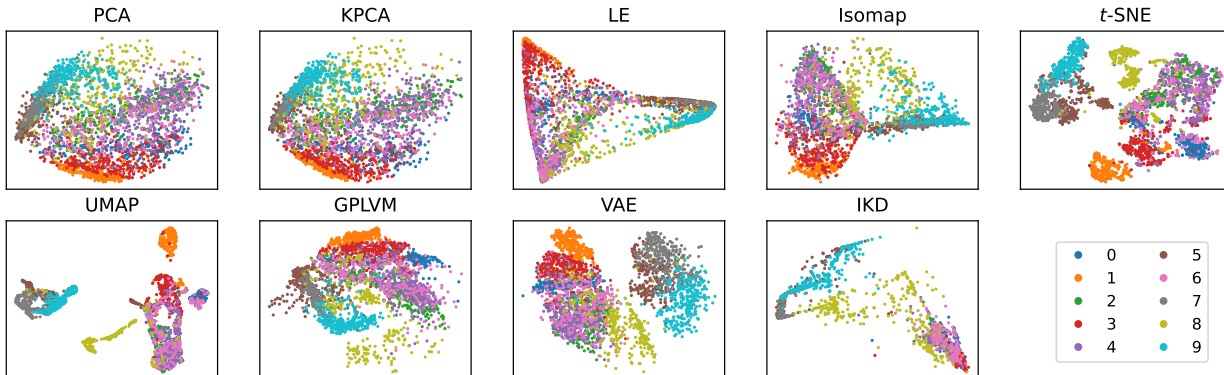

Figure 11: Visualization of the dimensionality reduction results of different methods on the F-MNIST dataset.

**Dimensionality reduction results of IKD with different kernels.** As we derived in Eq. 10 in the main content, choosing different kernels does not affect the dimensionality reduction results too much in terms of the latent structure similarity since different kernels have similar shapes. Especially in a classification task, although the exact relationship between the estimated latent data points under different kernels is not the same, points belonging to the same class consistently locate together in latent space (low dimensional space) no matter what kernel is used. Fig. 12 shows that the dimensionality reduction results under different kernels are nearly the same (up to reflection, rotation, and translation). Tab. 2 also shows similar 5-fold cross-validation $k$-NN classification accuracies under different kernels.

**Varying number of points $T$.** Although a theoretical analysis of the time complexity of IKD w.r.t. the number of data points $T$ has been provided in Sec. 2.4 and discussed in Sec. 3.1, we are still curious about the comparisons of different methods on datasets varying number of data points $T$. Therefore, we apply

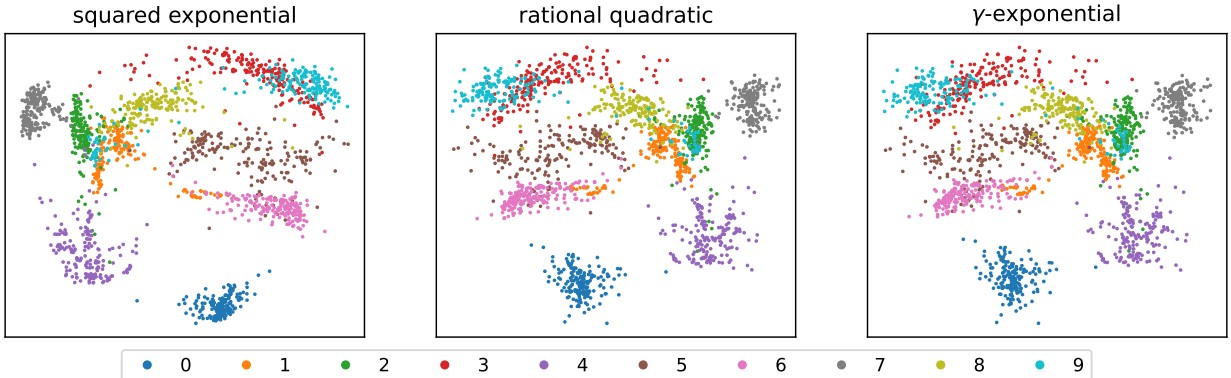

Figure 12: Dimensionality reduction results of IKD with different kernels for the digits dataset with the latent dimensionality $M = 2$. $\alpha = 1$ in the rational quadratic kernel, and $\gamma = 1$ in the $\gamma$-exponential kernel.

Table 2: 5-NN 5-fold cross-validation classification accuracies under different kernels and latent dimensionalities $M \in \{2, 3, 5, 10\}$ for the digits dataset.

| $k$ | kernels | 2 | 3 | 5 | 10 |
|---|---|---|---|---|---|
|  | squared exponential | 0.875899 | 0.85085 | 0.946049 | 0.944937 |
| 5 | rational quadratic | 0.841382 | 0.821323 | 0.931574 | 0.935474 |
|  | $\gamma$-exponential | 0.837478 | 0.806288 | 0.930458 | 0.933807 |
|  | squared exponential | 0.872006 | 0.844732 | 0.936592 | 0.937696 |
| 10 | rational quadratic | 0.857527 | 0.825235 | 0.92212 | 0.943258 |
|  | $\gamma$-exponential | 0.854737 | 0.81242 | 0.919336 | 0.93992 |
|  | squared exponential | 0.871453 | 0.843067 | 0.928804 | 0.932683 |
| 20 | rational quadratic | 0.857521 | 0.822467 | 0.906541 | 0.929341 |
|  | $\gamma$-exponential | 0.856408 | 0.817457 | 0.908767 | 0.928231 |

different methods on the F-MNIST dataset with the number of data points $T \in \{1000, 2000, 3000, 4000\}$. The $k$-NN accuracies and silhouette scores in Fig. 13(a) and (b) show that IKD is consistently the best among all eigen-decomposition-based methods, except for the silhouette score at $T = 4000$. In terms of runtime, we can see that IKD is faster than optimization-based methods. However, comparing the growth rate of IKD with UMAP and VAE, we can see from the growth rates shown in Fig. 13(c) that, due to (1) the worse time complexity of IKD and other eigen-decomposition-based methods and (2) the good scalability of optimization-based methods to large-scale datasets, the impact of the number of data points $T$ to IKD and other eigen-decomposition-based methods gets bigger and bigger as $T$ increases.

## 4    Discussions

In summary, IKD, as an eigen-decomposition-based method, consumes a short running time but is able to obtain dimensionality reduction results better than other eigen-decomposition-based methods. When facing high-dimensional observation data, IKD can perform significantly better than all other methods in a very short time.

Note that although eigen-decomposition-based methods perform relatively worse than optimization-based methods, the benefit of fast running provides good initialization for sophisticated nonlinear optimization problems, mitigating the numerical instability and multi-modal issues commonly observed in methods such as GPLVM and VAE.

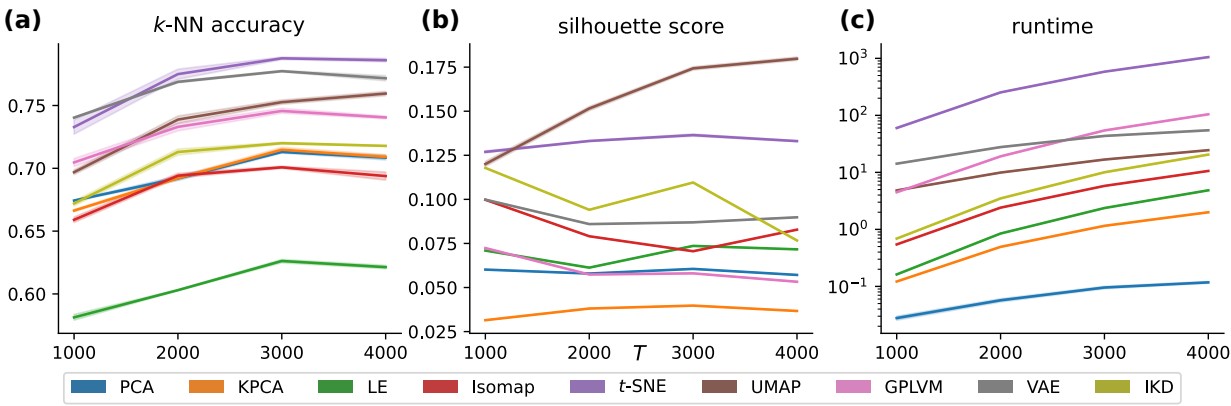

Figure 13: (a) average $k$-NN 5-fold cross-validation, (b) silhouette score, and (c) running time, by different methods, on the F-MNIST dataset including different numbers of data points $T$.

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
