# OpenReview forum: "Inverse Kernel Decomposition"
_TMLR — Accepted by TMLR_

### Review · Reviewer_Mh8Z · 2023-11-18

**Summary Of Contributions:**

This paper proposes inverse kernel decomposition (IKD), a dimensionality reduction method based on the Gaussian process latent variable model. The proposed method is simple and efficient. Promising experimental results are reported on synthetic and real data.

**Audience:**

Yes

**Claims And Evidence:**

Yes

**Requested Changes:**

It is suggested to revise the manuscript according to the above Weaknesses.

**Strengths And Weaknesses:**

Strength:

1. The proposed inverse kernel decomposition seems new to the best of my knowledge.

2. The proposed method is simple and efficient.

3. The empirical performance is promising.

Weaknesses:

1. Given so many eigen-decomposition-based dimensionality reduction method, the novelty of this paper seem rather limited.

2. The motivation is not clearly stated. Why do the authors need to rely the dimensionality reduction method on the Gaussian process latent variable model?

3. The writing is not always satisfactory. For example, there are typos like "all pairs of rows in Z. I.e., ..."

---

### Review · Reviewer_9zQM · 2024-01-02

**Summary Of Contributions:**

This paper focuses on dimensionality reduction problems. The authors propose a novel nonlinear dimensionality reduction method, called as Inverse kernel decomposition (IKD), for this purpose. Several numerical simulations have been conducted to illustrute the effectiveness and outperformance of IKD.

**Audience:**

Yes

**Claims And Evidence:**

Yes

**Requested Changes:**

1. The paper should be rewritten, especially to highlight the motivation (why IKD performs well rather than exhibiting tedious mathematical expressions), algorithm itself (how the algorithm implementation, why the specific kernels are adopted) and the explanations of the algorithms (the effect of algorithmic parameters, the computation burden, etc..).

2. Highlight the problem setting, especially demonstrating how to measure the quality of the mentioned approaches to show the advantage of the proposed IKD.

3. Experiment: 1) The effect of noise, or the outperformance of IKD, should be added.
                         2) The selection of algorithm parameters (or kernels) and the role of the algorithm parameters should be given.

**Strengths And Weaknesses:**

Strengths: The numerical results, especially for real data experiments are good, demonstrating the advantages of IKD, compared with PCA, KPCA, LE, Isomap, t-SNE, UMAP, GPLVM and VAE.

Weaknesses:

1. The paper is not written in a good shape. For example, there lacks several words to introduce the ``problem setting''. In particular, how to measure the quality of a dimensionality reduction is not presented.

2. The authors argue that their approach is capable of tackling noisy data. However, in their simulations, the noise is too small, i.e. \mathcal N(0,0.05^2).

3. The motivation and algorithm are not written well. There are so many tedious but simple mathematical computations that are easy to derive.

---

> ### Author Response · Authors · 2024-01-21
> **Reply to Reviewer 9zQM**
>
> Dear Reviewer 9zQM,
>
> Thank you very much for your time and valuable comments on our paper. We highly appreciate your recognition of the strengths of our paper.
>
> ### Weaknesses and Requested Changes
> * **Problem setting and measure the quality of a dimensionality reduction result**:
>     * We have highlighted the problem setting in Sec. 2.1 in our revision.
>     * For synthetic datasets, we align the estimated latent with the ground truth through an affine transformation (i.e., a linear decoder) and compute the $R^2$ metric.
>     * For real-world datasets, there is no ground truth latent. Since there are class labels for those datasets, we hope the dimensionality reduction result could separate data points with different labels and cluster data points with the same labels. Therefore, we use $k$-NN decoders to report the classification accuracy based on the learned latent. As suggested by Reviewer jwfe, we also report the silhouette score based on the learned latent. These evaluations have been highlighted in our revision (i.e., the 2nd paragraph of Sec 3.1 and the 2nd paragraph of Sec 3.2).
> * **Noise**: Thanks for this point, we have added a new experiment varying the noise level to see how IKD performs with different noise levels (Fig. 5) in our revision.
> * **Motivation and algorithm**:
>     * We would like to note that our math derivation is not purely solving a matrix eigen-decomposition, but aims to solve the Gaussian process latent variable model (GPLVM), and this eigen-decomposition solver (IKD) is not trivial to people working with GPLVM. In other words, it is our contribution that we show that we can use such a simple eigen-decomposition method to solve the complicated GPLVM problem. Therefore, we think it is worth showing how to solve the GPLVM problem step-by-step.
>     * Sec. 2.3 shows how to adapt the algorithm with different choices of the kernel, which is dependent on the kernel used in the generative process of GPLVM.
>     * Sec. 2.4 analyzes the time complexity of the algorithm.
> * **Selection of algorithm parameters**:
>     * Thanks for this point. We have highlighted that we will use the most commonly used square exponential kernel as the default kernel in our latest revision.
>     * For algorithm parameters of other baseline methods for comparison, we have reviewed and updated their parameter details at the beginning of Sec. 3 in our latest revision.
>
> We hope that this response addresses the majority of your concerns and questions. We have made changes according to your suggestions (highlighted by yellow in our latest revision). Your feedback is valuable, so please don't hesitate to provide additional comments or ask further questions.

---

> > ### Comment · Reviewer_9zQM · 2024-01-23
> >
> > All my concerns have been successfully settled. I have no further comments

---

> > > ### Comment · Reviewer_9zQM · 2024-02-09
> > > **Accept**
> > >
> > > I have no further comments

---

### Review · Reviewer_jwfe · 2024-01-08

**Summary Of Contributions:**

This article presents a dimension reduction method based on estimating latent variables from a kernel derived from observations. The core idea is to see the empirical covaraince matrix as an estimation of a kernel matrix between latent variables (under Gaussian assumption) and to try to estimate this latent variables with eigenvalue decompostion. The original idea of considering the intermediate step of a Gaussian process for dimension reduction is intersting a seems to work reasonably well in practice.

**Audience:**

Yes

**Broader Impact Concerns:**

No ethical implications of the work.

**Claims And Evidence:**

No

**Requested Changes:**

Improve the experiments as detailed above + provide more clarity in establishing connections with previous standard spectral methods for dimension reduction.

**Strengths And Weaknesses:**

Strengths:
- The method is simple and easy to implement
- It has interesting connections with spectral methods
- It works reasonably well in practice

Weaknesses:
- Connections to prior standard methods
- Experiments do not support the different claims

In my opinion, there are significant connections to standard methods that are missing, especially regarding the proposed method and MDS, which do not seem to be sufficiently discussed (only in 2.5). In fact, the proposed method seems to do almost exactly what MDS would do, with a minor modification. For MDS, one observes measurements of 'distances' (in a broad sense) $\delta_{ij}$ and seeks to find latent positions of a squared distance matrix (EDM) $d_{ij} = |x_i - x_j |^2$ such that $\delta_{ij} \approx d_{ij}$. To reconstruct the points from the set of distances, a very classical approach is based on the eigenvector decomposition of the covariance matrix (see e.g., [Section B, 1] or [3]). In fact, the approach proposed here is exactly the same approach but with a special choice of $\delta_{ij}$ that comes from a similarity kernel (mapped to a notion of ''distance'' with the $f^{-1}$). These important references are almost not discussed, as Part 2.2 is kind of presented as an original contribution, whereas it is a small modification of very classical algorithms (whose original ideas trace back to Gower in 1966 [2]). It seems to me that this method is actually doing a simple MDS but in the embedding space defined by the kernel. To make these connections clearer, I think it is important to derive the loss that is minimized by the algorithm.

Another point is that there is no guarantee that the presented approach actually leads to a good estimator of the true kernel $K$ compared to classical maximum likehood estimators (as in Gaussian processes). It would be really interesting to prove that the estimator proposed is somehow consitent and is able to estimate the true latent positions.

Moreover, a recurring claim in this article is the algorithm's speed, yet there is no analysis of algorithmic complexity. It doesn't seem that a solution based on eigenvalue decomposition is especially efficient, as it is of the order of $O(T^3)$ in computation. Concerning this point, for displaying runtimes, a logarithmic scale on the y-axis would provide a better idea of the differences between methods because, in Figure 2, they all appear roughly identical (except GPLVM). Additionally, for a fair comparison and to claim that "IKD is able to provide competitive performance against some SOTA optimization-based methods but at a much faster running speed" it would be necessary to calculate the speed of methods for varying $T$ and not just for a dimension $N$ that varies.

Finally, the experiments don't seem to support the claim that "As an eigen-decomposition-based method, IKD achieves more reasonable latent representations" and "IKD always obtains improved performance with an increasing observation dimensionality, and claims its absolute superiority when the observation dimensionality is very large." Indeed, the only metric proposed in the experiments is a classification metric with a $k$-NN on the representations. To evaluate the quality of an embedding and claim that "IKD always obtains improved performance" this doesn't seem sufficient to me. It's also important to compare methods with other metrics such as the silhouette score or trustworthiness. Additionally, Figure 6 is quite challenging to interpret, making it difficult to draw any conclusions. Variance is not presented, the curves are hard to read, and the significance of different $k$ values is unclear (the three lines are almost identical). Here, an aggregated table seems more appropriate, with a setting such as $k=1$ for example. Finally, the experimental setup for competing methods is not entirely clear, especially since t-SNE is sensitive to the perplexity parameter, and it's unclear how this parameter is chosen.

[1] Euclidean Distance Matrices. Essential Theory, Algorithms and Applications. Ivan Dokmanic, Reza Parhizkar, Juri Ranieri and Martin Vetterli.

[2] Gower, John C. Some distance properties of latent root and vector methods used in multivariate analysis, 1966.

[3] Multidimensional Scaling, Sammon Mapping, and Isomap: Tutorial and Survey, Benyamin Ghojogh, Ali Ghodsi, Fakhri Karray, Mark Crowley.

---

> ### Author Response · Authors · 2024-01-21
> **Reply to Reviewer jwfe (1/2)**
>
> Dear Reviewer jwfe,
>
> Thank you very much for your time and valuable comments on our paper. We highly appreciate your recognition of the strengths of our paper.
>
> ### Weaknesses
> * **Connections with previous standard spectral methods, i.e., Isomap and MDS**: Thank you very much for this valuable suggestion. We have added a paragraph (the 3rd paragraph in our current revision in Sec. 1) discussing the similarities and differences between our IKD and Isomap (MDS). Also, Isomap as a generalized version of MDS is one of the baseline methods to be compared in our experiments (Sec. 3).
> * **Guarantee of the estimated latent and the loss function of the algorithm**:
>     * From the derivation in Sec. 2.2, if the data is generated from GP, then as the number of observation dimensionality $N$ increases, we should get a correlation matrix with better quality. As $N\to\infty$, the sample covariance matrix $\boldsymbol S$ converges to the kernel covariance matrix $\boldsymbol K$, then the estimated latent $\tilde{\boldsymbol U}$ converges to the true latent $\boldsymbol Z$ up to an affine (linear) transformation.
>     * The loss that is minimized is Eq. 8, which can be directly solved by a one-step matrix decomposition. Note that the solution provided by Eq. 9 is a bit different from solving SVD, since we need to find its optimal rank-$M$ positive semi-definite approximation.
>     * To evaluate this step of eigen-decomposition solving the minimization problem, we added the explained variance ratio $\frac{\sum_{t=1}^M}{\lambda_t^2}{\sum_{t=1}^T\lambda_t^2}$ in our revision as a quantification.
> * **Speed**:
>     * Thanks for pointing this out. We agree that log-scale plots for running times are more appropriate and we have made these changes in our revision (Fig. 2(b,e,h), Fig. 4(c), Fig. 7(c), and Fig. 13(c)).
>     * Regarding the running time, we acknowledge that eigen-decomposition-based methods are strictly higher than $O(T)$ but optimization-based methods are just $O(T)$. In the 2nd to last paragraph of Sec. 3.1, we clarified this point that when increasing $T$, optimization-based methods can make use of stochastic gradient descent to accommodate large-scale datasets (i.e., a large number of data points $T$). In fact, all eigen-decomposition-based methods or methods recovering a $T\times T$ kernel suffer from this drawback. For fair comparison, we think extra scaling techniques should be incorporated to deal with large-scale eigen-decomposition, which falls out of the scope of this paper. Theoretically, we have shown in Sec. 2.4 that the blockwise solution of IKD is $O(MT^3)$ and the geodesic solution of IKD is $O(T^2\log(T))$.
>     * Although a theoretical analysis of the time complexity of IKD w.r.t. the number of data points $T$ has been provided in Sec. 2.4 and discussed in Sec. 3.1 as mentioned above, we do agree that adding experiments with varying numbers of data points $T$ is worthwhile. Therefore, we added a new experiment to the last paragraph in Sec 3.2. We apply different methods on the F-MNIST dataset with the number of data points $T\in\{1000,2000,3000,4000\}$. The $k$-NN accuracies and silhouette scores in Fig. 13 (a) and (b) show that IKD is consistently the best among all eigen-decomposition-based methods, except for the silhouette score at $T=4000$. In terms of runtime, we can see that IKD is faster than optimization-based methods. However, comparing the growth rate of IKD with UMAP and VAE, we can see from the growth rates shown in Fig. 13(c) that, due to (1) the worse time complexity of IKD and other eigen-decomposition-based methods and (2) the good scalability of optimization-based methods to large-scale datasets, the impact of the number of data points $T$ to IKD and other eigen-decomposition-based methods gets bigger and bigger as $T$ increases.
> * **Real-world datasets classification evaluation**:
>     * For the classification task, our use of $k$-NN as the decoder to evaluate the dimensionality reduction performance is the same as that used in the UMAP paper [https://arxiv.org/abs/1802.03426].
>     * We definitely agree that more metrics can make the results more convincing. Therefore, we have added the silhouette score as another metric (Fig. 7) in our current revision.

---

> > ### Author Response · Authors · 2024-01-21
> > **Reply to Reviewer jwfe (2/2)**
> >
> > * **Regarding Fig. 6 (now it is Fig. 7 in the latest revision)**:
> >     * In fact, we used tables in our initial draft. Then, we realized that tables are not able to reflect the increasing accuracy w.r.t. increasing latent dimensionality $M$. Furthermore, by these line plots, different methods on different datasets can be visually compared, so that we can see that IKD achieves very promising results on the COIL-20 dataset with high observation dimensionality $N$. Besides, there is no need to find differences between different $k$-NN results with $k\in\{5,10,20\}$, since trying different $k$ is a way to make sure we are not cherry-picking a particular $k$ that has good performance, but the results are consistent when varying $k$. Through this way, we can be convinced that it is because of the good quality of the dimensionality reduction result (i.e., the latent). Therefore, we finally chose line plots rather than tables, and hence, we kindly hold our perspective that line plots make the results more intuitive.
> >     * We have added error bars to Fig. 7 in our revision. However, the standard error is very small so most of them are not clearly visible. Also, note that there is no randomness for eigen-decomposition-based methods, so there is no error bar in Fig. 7(a) accuracy and Fig. 7(b) silhouette score for eigen-decomposition-based methods.
> > * **Hyperparameters details of competing methods**: Thanks, we have added these details to the beginning of Sec. 3.
> >
> > We hope that this response addresses the majority of your concerns and questions. We have made changes according to your suggestions (highlighted by yellow in our latest revision). Your feedback is valuable, so please don't hesitate to provide additional comments or ask further questions.

---

### Decision · Action_Editor_rEhv · 2024-02-29

**Recommendation:** Accept with minor revision

**Comment:**

The paper introduces a novel nonlinear dimensionality reduction method named Inverse Kernel Decomposition (IKD), which relies on an eigen-decomposition of the sample covariance matrix of data and draws inspiration from Gaussian process latent variable models. Reviewers have acknowledged the novelty of the proposed method. **Two reviewers**  have strongly recommended clear acceptance following the rebuttal with the authors.

**One reviewer** commended the authors' efforts in enhancing the manuscript during the rebuttal; however, they expressed some conservatism regarding the current version. Specifically, they suggested two points for improvement:

1) Enhancing the readability of the experiments by ensuring clarity in the figures and explanations of the conclusions.
2) Providing more extensive discussions on the connections with prior research and existing spectral methods. While the authors mention Isomap, it appears that their approach may be viewed as a specific instance of kernel PCA with a particular choice of kernel.

Considering these suggestions, I recommend acceptance with minor revisions. I encourage the authors to address the aforementioned comments to further strengthen the paper.

**Audience:**

Nonlinear dimension reduction remains a central topic in machine learning and statistics.  The proposed method is new which is inspired by Gaussian process latent variable models.  The paper is believed to be interesting to the TMLR audience.

**Claims And Evidence:**

Yes.  The method is new as clearly discussed in the reviewers' report and the effectiveness of the proposed is supported by experiments.